# Biomarkers and Management of Cholangiocarcinoma: Unveiling New Horizons for Precision Therapy

**DOI:** 10.3390/cancers17071243

**Published:** 2025-04-06

**Authors:** Naoshi Nishida

**Affiliations:** Department of Gastroenterology and Hepatology, Faculty of Medicine, Kindai University Osaka 589-8511, Japan; naoshi@med.kindai.ac.jp; Tel.: +81-72-366-0221

**Keywords:** cholangiocarcinoma, mutations, liquid biopsy, non-coding RNAs, tumor immune microenvironment

## Abstract

This review aims to provide a comprehensive update on the development of biomarker and therapeutic agents in cholangiocarcinoma (CCA). CCA is an aggressive malignancy with limited early detection methods, necessitating reliable biomarkers for diagnosis and management. Conventional tumor markers lack diagnostic accuracy; recent advancements in next-generation sequencing have identified actionable mutations, such as *FGFR2* fusions and *IDH1/2* mutations, enabling targeted therapies that improve survival. Cancer panels facilitating multiplex profiling reveal actionable mutations, with mutation-guided therapies markedly enhancing outcomes. Liquid biopsies, including cfDNA and ctDNA, offer non-invasive, real-time tumor monitoring. Non-coding RNAs in serum and bile also show promise as diagnostic and prognostic biomarkers. Immune checkpoint inhibitors (ICIs) demonstrate efficacy in subsets of patients; analyses of tumor immune microenvironment (TME) may also provide valuable information. Integrating multi-omics approaches and advanced technologies holds the potential to revolutionize precision medicine, enabling early detection, improved prognosis, and personalized therapies in CCA.

## 1. Introduction

The incidence of cholangiocarcinoma (CCA) has been increasing globally, and early detection and curative surgical resection are pivotal for improving the prognosis of patients with CCA. However, the progressive nature of CCA, characterized by a lack of early symptoms, often precludes timely diagnosis. Furthermore, there are currently no effective screening methods for high-risk populations, underscoring the pressing need for the identification of non-invasive biomarkers that can contribute to the detection of lesions and improved outcomes.

Currently, systemic therapy is the primary treatment for CCA patients with advanced stage and post-surgical recurrence. Conventional therapeutic agents, such as those involving gemcitabine, cisplatin, or tegafur–gimeracil–oteracil potassium (TS-1), have formed the backbone of systemic treatment for years (Table 1). However, advances in molecular oncology have unveiled actionable driver mutations in CCA, paving the way for the development and clinical application of molecular-targeted therapies tailored to specific genetic alterations. Moreover, significant strides have been made in the immunotherapy field, particularly with the advent of immune checkpoint inhibitors (ICIs). Combining ICIs with conventional cytotoxic agents such as gemcitabine and cisplatin (GC) has demonstrated promising efficacy and is increasingly being adopted in clinical practice for the treatment of CCA. Despite these advancements, the development of robust biomarkers to predict treatment response and guide therapeutic decision-making remains an area of unmet need.

This review aims to provide a comprehensive overview of the role of biomarkers in the management of CCA. We discuss the current state of biomarker development for CCA, encompassing their utility in diagnosis, prognosis, and therapeutic stratification. We also explore emerging trends and challenges in integrating biomarkers into clinical practice, highlighting their potential to shape the future of precision medicine for CCA treatment (Figure 1).

## 2. Serum and Plasma Biomarkers for Cholangiocarcinoma

CCA is often diagnosed based on elevated levels of hepatic and biliary enzymes and total bilirubin. So far, carcinoembryonic antigen (CEA) and carbohydrate antigen 19-9 (CA19-9) are commonly applied for the diagnosis and management of CCA in patients with an elevation of serum hepatic and biliary markers.

### 2.1. Conventional Serum and Plasma Biomarkers

The concomitant elevation of CA19-9 and CEA is associated with advanced-stage CCA [8]. However, these conventional tumor markers are not specific to CCA and are frequently negative in the early stages. Additionally, serum levels of these tumor markers are also elevated in benign conditions, such as in cases of obstructive jaundice, further limiting their specificity.

Therefore, these tumor marker levels are utilized as an adjunct in the diagnosis of the disease [9]. Previous meta-analysis showed that the diagnostic sensitivity of CA19-9 was 0.72 and the specificity was 0.84; diagnostic accuracy of CA19-9 tends to be the same in different control types [10] and elevated CA19-9 levels are more frequently observed in patients with advanced tumor stage and poor survival. It should be noted that blood types of Lewis antigen-negative lead to the false negative for this tumor marker even if they carry malignant tumors [11]. Immunohistochemical staining is valuable in the differential diagnosis of CCA. For instance, it has been reported that immunostaining for cytokeratin (CK) 7, CK19, CK20, and caudal type homeobox 2 (CDX2) is useful in distinguishing CCA from metastatic liver cancer [12,13]. CDX2 is a nuclear transcription factor encoded in the epithelial cells of the gastrointestinal tract from the duodenum to the rectum. It is known to be frequently positive in gastrointestinal cancers, particularly colorectal cancer, with a positivity rate of 84%. In contrast, the expression rate of CDX2 in CCA is significantly lower [14].

### 2.2. Serum and Plasma Biomarkers Under Development

A serum-soluble CK 19 fragment (CYFRA 21-1) has also recently emerged as a promising biomarker for intrahepatic CCA (iCCA) compared to CA19-9. Elevated levels of CYFRA 21-1 in patients with iCCA, as opposed to those with benign biliary diseases, have demonstrated notable diagnostic potential [15,16]. Matrix metalloproteinase-7 (MMP-7), a zinc-dependent endopeptidase involved in extracellular matrix remodeling, has also emerged as a candidate biomarker. Elevated serum MMP-7 levels have shown diagnostic value in distinguishing iCCA from non-malignant biliary obstructions in patients with obstructive jaundice [17]. Several matricellular glycoproteins involved in cell adhesion, migration, and survival are also reported as potential prognostic biomarkers [18]. For instance, osteopontin levels are significantly higher in patients with CCA compared to those without malignancies [19]. Preoperative serum osteopontin levels correlate with overall survival (OS) rates following tumor resection [19]. Interleukin-6 (IL-6) has also been proposed as a diagnostic and prognostic marker for biliary tumors. Elevated serum IL-6 levels can distinguish patients with CCA from those with HCC, metastatic colorectal carcinoma, and benign biliary diseases. IL-6 exhibited sensitivity and specificity rates of 73% and 92%, respectively, in differentiating patients with CCA from healthy individuals [20]. Additionally, mucins, glycoproteins essential for epithelial protection, have shown potential as biomarkers. Elevated serum mucin 5AC levels are associated with reduced OS, warranting further validation for clinical integration [21]. Extracellular vesicle (EV)-derived proteins are another promising avenue in CCA biomarker research. Recent case-control studies have indicated that a specific profile of serum EV-derived proteins could predict the risk of CCA development in patients with primary sclerosing cholangitis (PSC) [22]. EV-derived protein panels have also demonstrated potential in distinguishing CCA from hepatocellular carcinoma (HCC) and in providing prognostic insights [22].

### 2.3. Cancer-Associated Metabolites

Altered serum lipid and amino acid levels have been reported in patients with iCCA compared to those with benign biliary diseases [23]. Panels of cancer-associated metabolic alterations have demonstrated superior diagnostic performance over CA19-9 and α-fetoprotein in distinguishing iCCA from HCC, even at early stages [23]. These profiles are also valuable for differentiating distal CCA (dCCA) from pancreatic ductal adenocarcinoma, as well as for identifying malignant stenoses [24,25]. A panel of 10 plasma metabolites has also shown prognostic value in predicting post-resection outcomes, aiding in tailoring treatment strategies [26]. However, no single serum or plasma biomarker has exhibited absolute specificity for the diagnosis of CCA, underscoring the necessity of integrating these findings with clinical data and other biomarkers.

## 3. Genetic Aberrations in Cholangiocarcinoma

CCA is classified into iCCA, perihilar cholangiocarcinoma (pCCA), and dCCA, depending on its anatomical location. The iCCA is further subdivided into large-duct and small-duct types, each exhibiting distinct molecular characteristics. In addition, differences in etiology across regions contribute to variations in these genetic abnormalities. For example, PSC is a well-recognized risk factor for CCA, especially in Western countries, whereas liver fluke infection, hepatitis B virus (HBV), and hepatolithiasis are predominant risk factors in Southeast Asia. According to the reports for the genetic analysis of CCA, frequent mutations are identified in genes such as *TP53*, *KRAS*, *SMAD4*, *CDKN2A/2B*, and *ARID1A* [27,28,29,30,31]. Importantly, CCA is characterized by the unique association between pathological and molecular features. Recently, several molecularly targeted therapies have been developed based on specific mutations, making pathological diagnosis essential for selecting appropriate treatment options.

### 3.1. Etiologies, Pathologies, and Genetic Alterations in the Small-Duct Type of Intrahepatic Cholangiocarcinoma

The iCCA is further subclassified into small-duct type and large-duct type based on the pathological features. Small-duct type iCCA frequently emerges from chronic liver diseases such as viral hepatitis and metabolic-associated steatohepatitis (MASH), primarily occurs in the peripheral liver, and predominantly exhibits a mass-forming (MF) growth pattern. Pathologically, it is characterized by small, low-grade tubular structures composed of cuboidal tumor cells with poor mucin production. Genetically, *FGFR2* fusions/rearrangements and *IDH1/2* mutations are frequently observed, making targeted therapy against these alterations feasible. *BAP1* and *ARID1A* mutations are also common. Notably, *IDH* mutations lead to the accumulation of 2-hydroxyglutarate, which induces epigenetic abnormalities. We have demonstrated a correlation between *IDH* mutations and aberrant methylation of the genes involved in antigen-presentation, leading to their downregulation [32,33]. These findings suggest that *IDH* mutations may contribute to the formation of an immunosuppressive tumor microenvironment (TME) through the downregulation of antigen-presenting machineries.

### 3.2. Etiologies, Pathologies, and Genetic Alterations in the Large-Duct Type of Intrahepatic, Perihilar, and Distal Cholangiocarcinoma

Conversely, the large-duct type iCCA shares etiological associations with pCCA/dCCA; PSC, hepatolithiasis, and liver fluke infection are their risk factors. The large-duct type iCCA predominantly arises near the hepatic hilum and exhibits either a periductal infiltrating (PI) growth pattern or a mixed MF+PI pattern. Clinically, it is characterized by rapid progression, frequent lymph node metastasis, and perineural invasion. Pathologically, it consists of columnar cells forming glandular structures with prominent mucin production, accompanied by desmoplastic fibrosis and a robust stromal reaction. Genetic alterations commonly observed in this subtype include *KRAS*, *TP53*, and *SMAD4* mutations, resembling the molecular profile of pCCA/dCCA and pancreatic cancer. The large-duct type iCCA exhibits strong inflammatory responses and high PD-L1 expression, suggesting the potential efficacy of ICIs. Similarly, pCCA, dCCA, and gallbladder cancer are characterized by predominant *TP53* and *KRAS* mutations, extensive fibrosis, and an inflammatory TME. *PKA* and *HER2* mutations were more common in dCCA. On the other hand, alterations in *EGFR*, *HER2*, and *ERBB3* were frequently observed in gallbladder cancer. Wardell et al. analyzed the genetic profiles of 412 CCA cases [30] and identified germline mutations in DNA repair genes, including *BRCA1/2*, *MLH1*, and *MSH2*. Jusakul et al. analyzed 489 CCA cases from different regions [29] and found that liver fluke-associated CCAs frequently exhibited *ERBB2* amplification and *TP53* mutations, whereas non-fluke-associated cases demonstrated a higher prevalence of copy number alterations, elevated PD-1/PD-L2 expression, *IDH1/2* and *BAP1* mutations, and *FGFR* rearrangements with epigenetic dysregulation.

### 3.3. Cancer Panels and Companion Diagnostics for Cholangiocarcinoma

In genetic analyses of CCA, actionable genetic alterations were detected in approximately 40% of cases [27]. Reports utilizing the FoundationOne™ platform, a comprehensive genomic profiling tool, have also provided valuable insights. *FGFR* alterations and *IDH1/2* mutations were predominantly detected in iCCA, with *FGFR2* alterations notably associated with younger age, female gender, and better prognosis [28]. A multivariate analysis examining the prognosis of patients with iCCA revealed that *TP53* mutations were associated with poor prognosis, whereas *FGFR* alterations were associated with favorable outcomes. Umemoto et al. analyzed 3031 cases of CCA using the FoundationOne™ platform [31] and reported a high frequency of *ERBB2* amplification in tumor mutation burden (TMB)-high tumors. Furthermore, *CDK12* rearrangements were predominantly identified in *ERBB2*-amplified CCAs. Among CCA cases diagnosed in individuals under 40 years of age, *GATA6* amplification, as well as *BRAF* and *FGFR2* rearrangements, were observed. Importantly, molecular-targeted therapies based on genetic abnormalities improved the prognosis of patients with iCCA, and molecular findings underscore the potential for actionable mutations to serve as therapeutic targets.

## 4. Molecular-Targeted Therapies and Biomarkers

CCA is characterized by multiple driver gene mutations, with mutation frequencies varying according to the tumor’s anatomical location. Genetic testing in oncology includes companion diagnostics and gene panel testing, which identifies multiple driver mutations and facilitates appropriate therapeutic strategies. Gene panel testing has shown promise in the clinical management of CCA [34]. Reports indicate that 68% of CCA cases can be treated based on genetic abnormalities, with the selection of targeted treatments in 53% of cases. Notably, patients receiving treatments guided by genetic abnormalities demonstrated a significantly improved OS compared to those treated without genetic guidance. Furthermore, utilizing treatments categorized as Clinical Actionability of Molecular Targets (ESCAT) I-II in the European Society for Medical Oncology (ESMO) Scale improved the prognosis of patients with iCCA [35]. Based on these findings, next-generation sequencing (NGS)-based gene panel testing using tumor tissue is recommended for CCA management [36]. So far, several studies have reported promising results for the treatment with molecular-targeted agents as detailed below (Table 2).

### 4.1. FGFR2 Gene Fusions/Rearrangements

The fibroblast growth factor receptor (FGFR) family comprises four receptors, FGFR1–4. Ligand binding to FGFR activates downstream signaling pathways, including the JAK-STAT, RAS/MAPK, and PI3K/Akt/mTOR pathways. *FGFR2* gene fusions and rearrangements induce the constitutive activation of these signaling pathways. Rearrangements of *FGFR2* occur in approximately 3.6–7.4% of CCA cases and are predominantly observed in iCCA [59]. *FGFR2* gene fusions and rearrangements serve as a biomarker for FGFR2 inhibitors and have been incorporated into companion diagnostics, underscoring their clinical importance in guiding targeted therapy [60,61]. Pemigatinib, an oral FGFR1–3 inhibitor, demonstrated efficacy in a phase II clinical trial targeting cases with *FGFR2* fusion/rearrangement, showing an objective response rate (ORR) of 35.5%, a median progression-free survival (PFS) of 6.9 months, and a median OS of 21.1 months (Table 2) [42]. Similarly, another FGFR1-3 inhibitor, infigratinib, showed efficacy in patients with *FGFR2* gene fusions/rearrangements, achieving an ORR of 23.1%, a median PFS of 7.3 months, and a median OS of 12.2 months [40]. Additional data regarding a phase II trial of futibatinib have also been reported [38,62]. While pemigatinib and infigratinib are reversible ATP-competitive inhibitors targeting FGFR1–3, futibatinib, an irreversible inhibitor targeting FGFR1–4, demonstrated promising results in pretreated iCCA cases with *FGFR2* gene fusions/rearrangements, achieving an ORR of 42%, a median PFS of 9.0 months, and a median OS of 21.7 months (Table 2) [38]. Importantly, irreversible inhibitors have shown efficacy in cases with acquired resistance to reversible FGFR inhibitors.

### 4.2. Mutations in the IDH Gene

Isocitrate dehydrogenase (IDH) is an enzyme that catalyzes the conversion of isocitrate to α-ketoglutarate. Mutant IDH enzymes further convert α-ketoglutarate to 2-hydroxyglutarate (2-HG), which promotes tumorigenesis by inducing epigenetic changes and impairing DNA repair. *IDH1* mutations are detected in approximately 20% of iCCA cases. Our research has revealed an association between *IDH1* mutations and methylation of antigen-presentation-related genes, leading to their decreased expression and the development of an immune-cold tumor immune microenvironment (TME) in CCA [32,33]. Therefore, increased serum levels of 2-HC can be a surrogate marker for *IDH1/2* mutation [63].

Ivosidenib is a small-molecule inhibitor targeting mutant *IDH1*, and its efficacy has been demonstrated in a phase III clinical trial (Table 2) [45]. The PFS was significantly prolonged in the ivosidenib group (median PFS of 2.7 months) compared to the placebo group (1.4 months). The median OS was 10.3 months in the ivosidenib group vs. 7.5 months in the placebo group. Notably, after adjusting for crossover, a significant difference in OS was observed between the two groups. Based on these findings, ivosidenib has been approved by the U.S. Food and Drug Administration (FDA) and is recommended as a second-line treatment for *IDH1*-mutant CCA.

### 4.3. Activating Mutations in the KRAS and BRAF Genes

Activation of the RAS/MAPK pathway is a common feature across various cancer types. In CCA, *KRAS* mutations are found in 9–40% of cases, with a higher prevalence in iCCA. Mutations in *BRAF*, particularly the V600E variant, have been identified as actionable targets. A phase II clinical trial investigating the combination of the BRAF inhibitor dabrafenib and the MEK inhibitor trametinib has shown promising results in pretreated patients carrying the *BRAF* V600E mutation (Table 2) [53]. Although this trial included multiple rare cancers with *BRAF* V600E, the ORR ranged from 46% to 53% for biliary tract cancers. This combination therapy is also recommended for *BRAF* V600E-mutant CCA.

### 4.4. HER2 Gene Amplification/Overexpression

Amplification or overexpression of *HER2*, a receptor tyrosine kinase involved in cell proliferation, is observed in 3–19% of CCA cases, with a higher prevalence in gallbladder cancer. HER2-targeted therapies are recommended for HER2-positive CCA in the NCCN guidelines. In a phase II trial evaluating the combination of pertuzumab and trastuzumab in HER2-positive previously treated CCA cases, this combination therapy achieved an ORR of 23.1%, a median PFS of 4.0 months, and a median OS of 10.9 months (Table 2) [58]. Additionally, the irreversible HER1, HER2, and HER4 inhibitor and anti-HER2 antibody has demonstrated efficacy in phase II clinical trials [56,57]. Other therapies, such as antibody–drug conjugates combining trastuzumab and the topoisomerase inhibitor deruxtecan, have also shown favorable outcomes in HER2-activated tumors [55].

### 4.5. Other Biomarkers for Molecular-Targeted Agents

The efficacy of pembrolizumab, an anti-PD-1 antibody, has been validated in the phase II KEYNOTE-158 trial for solid tumors exhibiting microsatellite instability-high (MSI-high) or mismatch repair deficiency (dMMR) [64,65]. However, the frequency of CCA with MSI-high or TMB-high is approximately only 2%. On the other hand, it is reported that 27.5% of biliary tract cancers carry genetic alterations within the genes involved in homologous recombination, suggesting that platinum agents or poly (ADP-ribose) polymerase inhibitors may be promising agents (Table 2) [66]. Neurotrophic receptor kinase (NTRK) fusion-positive solid tumors can be treated effectively with entrectinib and larotrectinib, although the prevalence of NTRK gene fusions in CCA cases is low [67,68].

## 5. Emerging Biomarkers and Their Future Perspectives

### 5.1. Tumor Cells and Cell-Free DNA in Peripheral Blood

The detection of tumor cells in the blood (circulating tumor cells: CTCs) has been reported, yet their clinical application remains an ongoing challenge due to the difficulty in isolating a single CTC among blood cells. The detection of tumor cells generally relies on epithelial markers, including CKs and epithelial cell adhesion molecules, while excluding leukocyte markers such as CD45. However, the heterogeneity of CTCs and variability in their biomarker expression, particularly during epithelial-to-mesenchymal transition, complicate the detection of cancer cells in blood. This variability underscores the need for multipronged approaches to detect and characterize CTCs effectively. Characterization of CTCs is a key strategy for utilizing liquid biopsies in diagnosis, monitoring of treatment efficacy, and disease surveillance [69].

In CCA, higher CTC counts have consistently been associated with poor prognosis. A previous study reported that CTC was detected in 17% of patients with CCA in 7.5 mL of their blood, which correlated with tumor extension and poor prognosis [19,21,70,71,72,73,74,75,76]. Baseline CTC detection was linked to reduced OS in nonresectable cases. Notably, similar detection rates of CTCs were observed across two clinical trials despite differences in sample volume and methodology [54,77]. CTCs are not merely considered prognostic markers but also substrates for metastasis, contributing to disease progression. In metastatic CCA, even the detection of as few as 1–3 CTCs correlated with poorer OS [78]. Although most CTCs do not survive in circulation [69], recent techniques have enhanced the sensitivity of detection [79]. Combining multiple affinity reagents, such as heparan sulfate-based probe SCH45, nucleic acid aptamers, and anti-EpCAM in microfluidic systems, has enabled the detection of CTC in nearly 100% of bile samples in CCA cases [80]. Furthermore, combining central and peripheral venous sampling has also improved detection rates from 40% to 54% [78]. These innovations highlight the growing importance of CTCs in precision oncology.

On the other hand, circulating nucleic acids, including cell-free DNA (cfDNA) and circulating tumor DNA (ctDNA), have gained significant attention for their potential in cancer management. In CCA, cfDNA and ctDNA analyses have shown promise for both diagnostic and prognostic applications. The evaluation of ctDNA offers several advantages over traditional tissue biopsies, including reduced invasiveness and the ability to capture tumor heterogeneity [81,82]. ctDNA carries tumor-specific genetic alterations, making it an invaluable tool for liquid biopsy. In advanced CCA cases, ctDNA was detectable in approximately 90% of patients, with variant allele fractions correlating strongly with tumor burden and stage of the disease [83,84]. Although the detection of ctDNA is challenging in the early stage, advancements in the detection of ctDNA will improve its sensitivity [85]. cfDNA testing also provides insight into mutational burden, dynamic tumor evolution, and acquired resistance mechanisms, aiding in patient stratification and therapy selection [73,86,87,88,89,90]. The prognostic significance of ctDNA also lies in its ability to reflect tumor dynamics and therapeutic responses. While ctDNA analyses have shown high sensitivity and specificity in detecting advanced CCA, most studies have focused on patients with established malignancies. Thus, early detection remains a critical challenge, particularly in identifying high-risk populations [91]. The detection of ctDNA carrying altered methylation is also reported as a complementary strategy, enhancing diagnostic accuracy and guiding clinical management [92,93,94]. Achieving concordance between mutations identified in ctDNA and tumor tissues is essential for validating cfDNA testing. Notably, better concordance has been observed in iCCA compared to dCCA, possibly due to differences in tumor location and patterns of cfDNA release [88]. Detecting gene fusions, such as *FGFR2* fusions in iCCA, can also be useful for the diagnosis of CCA because these alterations could be a driver for the development of CCA. However, current liquid biopsy panels face challenges for the detection of fusion genes and rearrangement, and innovative platforms beyond next-generation sequencing may help address these limitations [89,95].

Liquid biopsies using cfDNA and ctDNA hold significant promise for capturing the intrapatient molecular heterogeneity, allowing for real-time monitoring of tumor progression, assessment of therapeutic responses, and detection of minimal residual disease after surgery [91]. As technologies continue to evolve, the integration of liquid biopsy data with imaging and pathology could lead to a more comprehensive and dynamic understanding of cancer status. These advancements underscore the potential for liquid biopsies to revolutionize the clinical management of CCA, paving the way for earlier diagnosis, improved prognostication, and personalized treatment strategies. Combining these methods with cutting-edge technologies, such as single-cell sequencing, microfluidic systems, and methylation profiling, could significantly enhance their clinical utility.

### 5.2. Non-Coding RNAs

Non-coding RNAs (ncRNAs) are a diverse group of RNA molecules that do not code for proteins but play critical roles in various biological processes. These include microRNAs (miRNAs), long non-coding RNAs (lncRNAs), and circular RNAs (circRNAs). Recent studies have demonstrated that miRNAs, lncRNAs, and circRNAs are involved in regulating cell proliferation, invasion, metastasis, and the development of drug resistance in CCA, highlighting their potential as diagnostic and prognostic biomarkers (Figure 2) [96,97].

#### 5.2.1. MicroRNAs

Various studies have shown that miRNAs are associated with the response to systemic chemotherapy in CCA. For instance, the overexpression of miR-21 in CCA has been reported to downregulate the tumor suppressor gene phosphatase and tensin homolog (PTEN) and to be associated with the acquisition of gemcitabine resistance [98]. Elevated levels of serum miR-200c-3p have been linked to increased recurrence rates and decreased OS following surgical resection in CCA patients [99]. Similarly, plasma miR-183-5p, which is known to promote immunosuppressive macrophages and upregulate programmed death-ligand 1 (PD-L1) expression, has been associated with higher recurrence rates and worse OS in patients with iCCA [74]. By leveraging the stability of non-coding RNA in body fluids and their role in critical biological processes, these markers offer a minimally invasive approach for improving the management of CCA. For example, bile-derived miR-200c-3p was found to be more abundant in CCA patients compared to individuals with biliary obstruction caused by gallstones [72].

Regarding the role of miRNA as biomarkers for the response to systemic therapy, the miR-130a-3p may promote resistance to gemcitabine by inhibiting the expression of peroxisome proliferator-activated receptor gamma (PPARG), which is involved in nucleotide metabolism [100]. Additionally, miR-200b, miR-210, let-7a, and miR-181c, among others, have been shown to contribute to gemcitabine resistance by inhibiting apoptosis. Furthermore, miR-1249 and miR-200b/c might contribute to gemcitabine and 5-FU resistance through the induction of CD^+^133 cells, which are thought to be associated with cancer stem cells [101,102]. MiR-125a-5p is upregulated in CCA compared to normal tissues and is associated with decreased sensitivity to gemcitabine. Several other tumor-derived miRNAs have been identified as playing important roles in gemcitabine resistance in CCA [103,104,105,106].

5-FU is another drug commonly used in the treatment of CCA, and several studies have clarified the role of miRNAs in mediating 5-FU resistance. Overexpression of miR-20a-5p has been found to suppress 5-FU resistance induced by the onco-lncRNA FALEC and to promote apoptosis [107]. On the other hand, miR-328 has been reported to inhibit tumor growth and to enhance apoptosis induced by 5-FU [108]. MiR-200b/c inhibits tumor migration and enhances the sensitivity of CCA cells to 5-FU [101]. MiR-885-5p may increase the sensitivity of CCA cells to 5-FU by targeting myotrophin (MTPN), a protein associated with cell migration and invasion [109].

Platinum-based chemotherapy, particularly cisplatin in combination with GC, is currently the standard treatment for advanced CCA. However, the development of cisplatin resistance remains a significant challenge in CCA therapy. MiR-199a-3p is an miRNA associated with cisplatin resistance in various types of tumors, including CCA. The upregulation of miR-199a-3p has been shown to enhance CCA cell sensitivity to cisplatin by inhibiting the expression of multidrug resistance protein 1 [110]. MiR-637 is involved in drug resistance in various cancers, and its downregulation in CCA is associated with resistance to cisplatin. Furthermore, miR-520c-3p has been reported to induce the suppression of cancer stemness and epithelial–mesenchymal transition (EMT), thereby enhancing CCA cell sensitivity to cisplatin [111].

#### 5.2.2. LncRNAs

Long non-coding RNAs (lncRNAs) are non-coding RNA molecules longer than 200 nucleotides and have been implicated in both carcinogenesis and tumor suppression [112]. Numerous lncRNAs localized in exosomes have been secreted into plasma, urine, and bile as highly tissue-specific circulating RNAs, which may serve as non-invasive biomarkers [113]. Several lncRNAs have been identified in serum and urine EVs from patients with CCA, PSC, and healthy controls. In serum-derived EVs, metastasis-associated in lung adenocarcinoma transcript-1 (MALAT1) and LOC100190986 showed high accuracy in distinguishing between CCA and PSC. In addition, LOC100134868 derived from urinary EVs demonstrated a high diagnostic value between CCA and healthy controls [113]. Most lncRNAs are significantly expressed in the tissues, cells, and bile of CCA and have diagnostic and prognostic potential as CCA biomarkers. Bile-associated some lncRNAs exhibited higher levels in CCA patients compared to control subjects with bile duct obstruction. Ge et al. discovered two lncRNAs, ENST00000588480.1 and ENST00000517758.1, to be highly expressed in exosomes derived from bile of CCA patients [114]. When combined for diagnosis, the area under the curve (AUC), sensitivity, and specificity were 0.709, 82.9%, and 58.9%, respectively, with sensitivity surpassing that of serum CA19-9. It was also reported that higher expression of these two lncRNAs in CCA patients was associated with poor survival, suggesting their potential as predictive markers for monitoring CCA [114]. On the other hand, lncRNA-NEF was downregulated in iCCA tissues, and its overexpression suppressed tumor cell migration and invasion by inhibiting runt-related transcription factor 1 (RUNX1), demonstrating good diagnostic characteristics to distinguish iCCA from healthy controls. Furthermore, low expression of lncRNA-NEF in iCCA patients was associated with significantly shorter OS, suggesting its potential as a biomarker [97]. Another study found that deleted in lymphocytic leukemia 1 (DLEU1) has been shown to correlate with advanced tumor lymph node metastasis (TNM stage) and can be used as a prognostic marker for CCA [115]. MALAT1 has been linked to poor prognosis in pCCA patients, with overexpression associated with decreased OS, worsened TNM stage, increased tumor size, and metastasis. Shi et al. reported that MALAT1 in plasma could serve as a useful diagnostic biomarker for pCCA [116]. Another study found that prostate cancer-associated transcript 1 (PCAT1) contributes to cancer progression via the Wnt/β-catenin signaling pathway and is elevated in extrahepatic CCA [117]. The overexpression of PCAT1 was associated with adverse outcomes in CCA patients, making it a useful prognostic marker.

H19 is upregulated in CCA tissues and is associated with tumor size, TNM stage, postoperative recurrence, and OS. It has been observed to have moderate sensitivity in distinguishing CCA tissues from normal tissues, with an AUC of 0.7422. Furthermore, when combined with other lncRNAs, the diagnostic sensitivity and specificity for differentiating CCA tissues from normal tissues increased [118,119]. Jiang et al. reported that the expression level of colon cancer-associated transcript (CCAT)1 in CCA tumor tissues was significantly higher compared to adjacent normal tissues. Increased expression of CCAT1 was associated with lower histological differentiation, lymph node invasion, and advanced TNM stage. Patients with overexpression of CCAT1 had significantly poorer OS, making CCAT1 an independent prognostic factor for CCA [120]. CCAT2 expression is elevated in CCA and inversely correlates with OS in CCA patients. Thus, CCAT2 holds practical value as a prognostic marker for these patients, with AUCs for OS and PFS of 0.702 and 0.715, respectively [121]. Zinc finger E-box binding homeobox 1 (ZEB1)-AS1 is also overexpressed in CCA and has been shown to promote tumor growth and metastasis in both in vivo and in vitro experiments. High expression of ZEB1-AS1 was associated with lymph node invasion, advanced TNM stage, and shortened survival [122].

#### 5.2.3. CircRNAs

Circular RNAs (circRNAs) are covalently closed single-stranded RNAs; they are typically produced from the intermediate exons of protein-coding genes [123]. Due to their tissue specificity and stability, circRNAs are less susceptible to degradation by ribonuclease R; they are considered promising candidates as cancer biomarkers [124,125]. The expression of circRNA Cdr1 is significantly upregulated in tumor tissues compared to adjacent normal tissues, and it is strongly associated with lymph node invasion, progression of the TNM stage, and postoperative recurrence. Furthermore, Cdr1 was identified as a novel independent prognostic biomarker for predicting overall survival (OS) in CCA patients, with a sensitivity of 83.3% and specificity of 58.3% [126]. Xu et al. found that levels of circ-CCAC1 were elevated in EVs derived from the bile and serum of CCA patients [127]. The diagnostic capability of circ-CCAC1 in serum was comparable to that of serum CA19-9, while in bile samples, it outperformed CA19-9. Interestingly, combining EV-derived circ-CCAC1 from bile or serum with CA19-9 enhanced diagnostic performance compared to either marker alone. The study also confirmed that high expression of circ-CCAC1 was an independent prognostic marker for iCCA and that circ-CCAC1 expression could predict postoperative recurrence in iCCA patients.

Additionally, hsa_circ_0000284 (cir-HIPK3) was reported to be upregulated in plasma exosomes from CCA patients [128]. Another study identified that hsa_circ_0020256 (cir-NSMCE4A) was highly expressed in exosomes secreted by tumor-associated macrophages and promoted proliferation, migration, and invasion of CCA cells [129]. This study also demonstrated a negative correlation between the expression of cir-NSMCE4A and both time to recurrence and OS in CCA patients. Moreover, high levels of hsa_circ_0030998 (cir-LAMP1) were associated with an increased rate of postoperative recurrence [130]. Hsa_circ_0003930 (cir-GGNBP2) was associated with a worse prognosis in patients after surgical resection. An increase in cir-GGNBP2 was reported as an independent risk factor for OS and RFS, suggesting its potential as a prognostic marker for iCCA [131]. Zhang et al. reported decreased expression of hsa_circ_0059961 (cir-ITCH) in CCA tissues [132]. Their study found a positive correlation between the expression of cir-ITCH and survival, with patients exhibiting high levels of cir-ITCH having better OS compared to those with low expression. Additionally, patients with high expression of hsa_circ_0008621 (cir-HMGCS1) in iCCA tissues had significantly shorter survival after radical resection and higher cumulative recurrence rates, indicating that cir-HMGCS1 may serve as an independent prognostic indicator for cumulative recurrence in iCCA patients [133].

## 6. Tumor Immune Microenvironment and Treatment Using Immune Checkpoint Inhibitors

CCA has demonstrated responsiveness to molecular-targeted agents (MTAs) as well as ICIs. MTAs focus on inhibiting driver mutations implicated in tumor progression, showing promise in advanced CCA cases resistant to first-line chemotherapy. Conversely, ICIs have also emerged as a transformative approach in CCA treatment. The TOPAZ-1 trial demonstrated that combining durvalumab (a PD-L1 inhibitor) with GC significantly improved OS, PFS, and response rates in unresectable or metastatic biliary tract cancer compared to chemotherapy alone (Table 3) [134]. Similarly, the KEYNOTE-966 trial validated the efficacy of combining ICIs with GC as first-line therapy in advanced CCA cases [135]. The TME plays a crucial role in determining the efficacy of ICIs, emphasizing the need to consider TME characteristics as potential biomarkers for CCA treatment.

Transcriptomic studies have classified the immune landscape of CCA into several subtypes of TME, reflecting their impact on immune evasion and patient outcomes, although there are still many challenges to overcome in developing biomarkers for the ICIs. Job et al. identified four immune subclasses—immune-desert, immunogenomic, myeloid, and mesenchymal—based on transcriptomic data from 198 iCCA samples [137]. The immune-desert subclass, lacking immune cell-related gene signatures, represented 46–48% of cases. Another classification by Martin-Serrano et al. divided CCAs into inflamed (35%) and non-inflamed (65%) classes and proposed the STIM (stroma, tumor, immune microenvironment) classification, which categorizes iCCA into five subtypes based on the composition of tumor cells, fibroblasts, and immune cells. [138]. Non-inflamed tumors included “hepatic stem-like”, enriched with *IDH1/2* and *BAP1* mutations as well as *FGFR2* fusions/rearrangements and characterized by abundant M2 macrophages; “tumor classical”, is associated with frequent aberrations in the cell cycle-related genes, *TP53* mutations, rapid progression, and poor prognosis; and “desert-like”, driven by Wnt/β-catenin signaling and a paucity of immune cells with predominance of regulatory T cells (Tregs). Inflamed tumors were subdivided into “immune classical”, characterized by metabolic pathway activation, extensive immune cell infiltration, such as abundant CD8+ T cells and γδ T cells, high PD-L1 expression; and “inflammatory stroma”, frequently harboring TGF-β activation and *KRAS* mutation, marked by extensive fibrosis, T cell exhaustion, and the presence of cancer-associated fibroblasts (CAFs). Based on these genetic and immune microenvironmental characteristics, it is possible that ICIs may be effective in the immune classical subtype, whereas KRAS inhibitors, FGFR or IDH inhibitors, and CDK inhibitors might be effective in the inflammatory stroma subtype, the hepatic stem-like subtype, and the tumor classical subtype, respectively. Zhu et al. further detailed spatial immunophenotypes of CCA, identifying inflamed, excluded, and ignored classes [139]. Inflamed tumors featured immune cell infiltration and activation of IFN-γ and IL-6/JAK/STAT pathways, making them potential targets for ICIs. In contrast, excluded and ignored phenotypes, defined by TGF-β, Wnt/β-catenin signaling, and angiogenesis, exhibited immune exclusion or evasion, possibly reducing ICI efficacy (Figure 3).

Most CCAs exhibit a non-inflamed TME and would be resistant to ICIs; thus, understanding mechanisms underlying immune evasion and resistance in CCA is crucial. The non-inflamed TME is, reportedly, associated with genetic alterations, such as *IDH1/2* mutations and *FGFR2* gene fusions, which suppress antigen presentation and foster an immunosuppressive microenvironment [33]. Therefore, these genetic alterations might be a biomarker for representing non-inflamed TME status, where ICI-based treatment may not be effective. As mentioned before, mutant IDH causes an increase of 2-HG; increased serum levels of 2-HC can be a potential surrogate marker for non-inflamed CCAs induced by *IDH1/2* mutation [63]. Thus, it is interesting to speculate that combining targeted therapies, such as IDH1/2 or FGFR inhibitors, with ICIs may restore anti-tumor immunity. Preclinical models suggest that these combinations can reinvigorate immune responses by reducing immunosuppressive signaling and enhancing antigen presentation [140,141].

Efforts to identify predictive biomarkers are critical for optimizing ICI use. Advances in multi-omics analyses and liquid biopsies may enable the precise classification of TME subtypes and the real-time monitoring of treatment responses. Additionally, integrating experimental models, such as organoids and single-cell technologies, with genomic and proteomic studies could unveil novel biomarkers and therapeutic targets.

## 7. Conclusions

CCA poses significant challenges in diagnosis and monitoring, differing from HCC, where many cases develop in cirrhotic livers under routine surveillance programs [142,143]. Over half of CCA cases occur in patients without underlying hepatobiliary diseases, which do not warrant inclusion in monitoring programs. The absence of high-risk group definitions and effective biomarkers for early detection complicates CCA surveillance efforts. Current research highlights the potential of liquid biopsy techniques, including bile-derived cfDNA and ctDNA, which detect molecular alterations in body fluid for early and accurate diagnosis [25,144]. Bile cfDNA, analyzed for mutations and methylation, is also a promising biomarker for the detection of CCA [144,145,146]. Additionally, combining liquid biopsy matrices with advanced technologies, such as single-cell analysis, could revolutionize CCA management [147]. Investments in research integrating experimental models and large patient cohorts are essential to developing novel biomarkers, offering hope for noninvasive and effective diagnostic strategies.

## Figures and Tables

**Figure 1 cancers-17-01243-f001:**
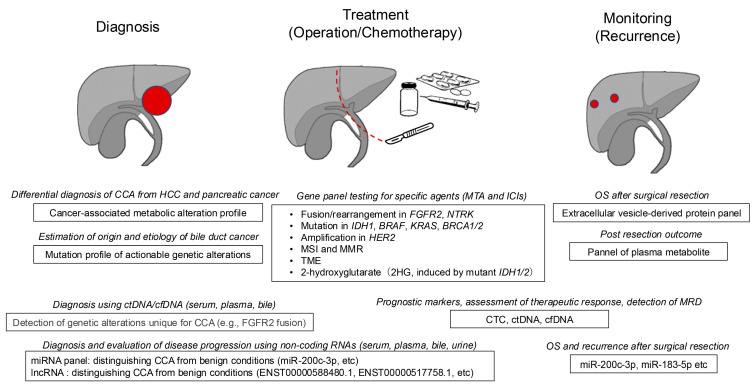
The candidate of biomarkers in the diagnosis, therapeutic selection, and monitoring of recurrence in cholangiocarcinoma. In addition to conventional serum biomarkers, novel biomarker candidates for the diagnosis, therapeutic selection, and monitoring of recurrence in cholangiocarcinoma (CCA) have been illustrated. Gene panel testing plays a critical role in determining the suitability of molecular targeted therapies for CCA. Moreover, biomarkers such as circulating tumor DNA (ctDNA), cell-free DNA (cfDNA), non-coding RNAs, extracellular vesicle-derived protein panels, and plasma metabolite panels have been reported for applications including diagnosis, evaluation of disease progression, and prognostic prediction. However, at present, these novel biomarkers, apart from gene panel testing, remain in the developmental stage.

**Figure 2 cancers-17-01243-f002:**
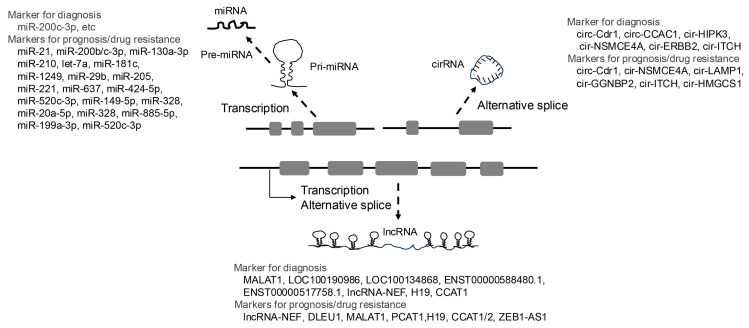
Development of non-coding RNA as biomarkers for cholangiocarcinoma. Non-coding RNAs (ncRNAs) are functional RNAs that are not translated into proteins. Among the various types of ncRNAs, the shortest are microRNAs (miRNAs), which are approximately 20–25 nucleotides in length. miRNAs function by binding to target mRNAs and inhibiting their translation. Primary miRNAs (pri-miRNAs) are processed through several sequential cleavage steps to produce mature miRNAs. In contrast, ncRNAs longer than 500 nucleotides are collectively referred to as long non-coding RNAs (lncRNAs). These include intronic, exonic, promoter-associated, and enhancer-associated lncRNAs, which are generated through transcription and alternative splicing. Similarly, circular RNAs (circRNAs) are predominantly derived from precursor mRNAs (pre-mRNAs) via alternative splicing. CircRNAs can be classified into exonic, exonic-intronic, and intronic circRNAs based on their composition. These non-coding RNAs are closely associated with the initiation, progression, and chemoresistance of cholangiocarcinoma (CCA) and hold promise as potential biomarkers for clinical applications.

**Figure 3 cancers-17-01243-f003:**
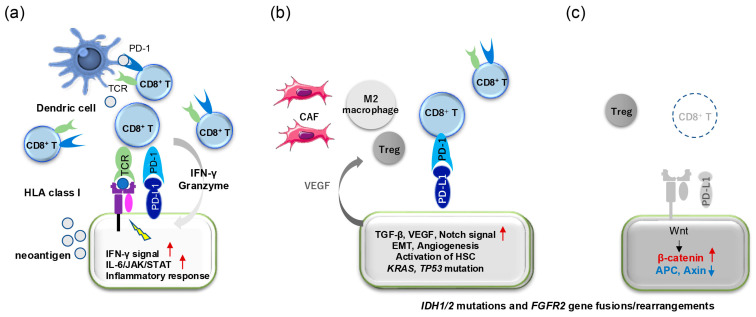
Schematic representation of the tumor immune microenvironment in cholangiocarcinoma. Several studies have classified the tumor microenvironment (TME) of cholangiocarcinoma (CCA). The red up arrows represent “activation” of the signal, and the blue down arrows show “inactivation”. In general, inflamed tumors exhibit the activation of anti-tumor immune responses, including interferon-gamma (IFN-γ) signaling, antigen presentation, and IL-6/JAK/STAT signaling. In such cases, immune checkpoint inhibitors (ICIs) may be effective (**a**). Conversely, a subset of CCA tumors displays an altered immune response characterized by increased TGF-β signaling, angiogenesis, and activation of hepatic stellate cells (**b**). In these tumors, various mesenchymal reactions—such as neovascularization, fibrosis, and epithelial–mesenchymal transition (EMT)—are induced, leading to restricted T cell infiltration. Additionally, the activation of TGF-β, vascular endothelial growth factor (VEGF), and Notch signaling promotes the infiltration of M2 macrophages and regulatory T cells (Tregs). In tumors lacking infiltrating immune cells, anti-tumor immune responses are largely absent, rendering ICIs potentially ineffective (**c**). The activation of the Wnt/β-catenin signaling pathway may contribute to this form of immune evasion.

**Table 1 cancers-17-01243-t001:** Phase II/III clinical trials of chemotherapy for cholangiocarcinoma.

Trial ID ^1^ (Study Name)	Phase (Line)	Agents	PEP ^2^	Patients	Results	Ref.
NCT04163900 (NuTide:121)	III (1st)	NUC-1031 + cisplatin vs. GC ^3^	OS, ORR	773 advanced BTCs patients	OS for NUC-1031/cisplatin vs. GC was. 9.2 months vs. 12.6 months. ORR for NUC-1031/cisplatin vs. GC was 18.7% vs. 12.4% (*p* = 0.049).	[1]
NCT03768414 (SWOG S1815)	III (1st)	GAP vs. GC	OS	441 advanced BTCs patients	GAP regimen to the standard GC did not improve OS. GAP vs. GC showed improvement in PFS among patients with GBC.	[2]
NCT03043547 (AIO NALIRICC)	II (2nd or later)	Nanoliposomal irinotecan, fluorouracil + leucovorin vs. fluorouracil + leucovorin	PFS	100 BTC patients with progression on gemcitabine-based therapy	Nanoliposomal irinotecan, fluorouracil + leucovorin did not improve PFS or OS.	[3]
jRCTs 031,180,082 ^4^	II (1st)	Oxaliplatin, irinotecan, leucovorin, fluorouracil	PFS	35 advanced or recurrent BTC patients	The median PFS and OS were 7.4 and 14.7 months. The study did not meet the PEP.	[4]
NCT01926236 (ABC-06)	III (2nd)	FOLFOX vs. symptom control	OS	162 BTC patients	OS was significantly longer in the FOLFOX group.	[5]
ChiCTR-TRC-14004733 ^5^	II (1st)	Gemcitabine + TS-1 vs. gemcitabine	ORR	62 advanced BTC patients	The ORR of the combination therapy and the monotherapy were 20.0 and 9.4%, respectively.	[6]
NCT00262769 (ABC-02)	III (1st)	CG vs. gemcitabine	OS	410 advanced BTC patients	OS = 11.7 months in the GC group and 8.1 months in the gemcitabine group (*p* < 0.001).	[7]

^1^ Trial IC registered in ClinicalTrials.gov. ^2^ PEP; primary endpoint. ^3^ NUC-1031; a phosphoramidate modification of gemcitabine. ^4^ jRCTs; Japan Registry of Clinical Trials. ^5^ ChiCTR; Chinese Clinical Trial Registry. Abbreviations: GAP; gemcitabine, nab-paclitaxel + cisplatin, GC; gemcitabine and cisplatin, OS; overall survival, BTC; biliary tract cancer, PFS; progression-free survival, GBC; gall bladder cancer, ORR; objective response rate, FOLFOX; folinic acid, fluorouracil, and oxaliplatin.

**Table 2 cancers-17-01243-t002:** Phase II/III clinical trials using molecular-targeted agents for cholangiocarcinoma.

Trial ID ^1^ (Study Name)	Phase (Line)	Agents ^2^	PEP ^3^	Patients	Results	Ref.
NCT02699606	II (2nd or later)	Erdafitinib	ORR	22 advanced CCA patients with *FGFR* alterations	ORR = 51.7% for patients with *FGFR* rearrangement 12.5% for patients with *FGFR* short variants.	[37]
NCT02052778 (FOENIX-CCA2)	II (2nd or later)	Futibatinib	ORR	103 unresectable iCCA patients with *FGFR2* fusion/rearrangement	ORR = 42%, with one CR and 42 PRs.	[38]
NCT02924376 (FIGHT-202)	II (2nd or later)	Pemigatinib vs. conventional systemic therapy	PFS	145 advanced CCA patients	PFS = 7.0 months for patients with *FGFR2* fusions/rearrangements in the pemigatinib group, which was longer compared with the systemic therapy group.	[39]
NCT02150967 (BGJ398)	II (2nd or later)	Infigratinib	ORR	108 patients with advanced CCA with *FGFR2* fusions/rearrangements	ORR = 23 · 1%, with one CR and 24 PRs.	[40]
NCT03656536 (FIGHT-302)	III (1st)	Pemigatinib vs. GC	PFS	Advanced CCA patients with *FGFR2* rearrangements	Ongoing.	[41]
NCT02924376 (FIGHT-202)	II (2nd or later)	Infigratinib	ORR	147 patients with advanced CCA with/without *FGFR2* alterations.	ORR = 35 · 5% in patients with *FGFR2* fusions or rearrangements (three CRs and 35 PRs).	[42]
NCT03773302 (PROOF 301)	III (1st; One gemcitabine-based therapy is permitted)	Infigratinib vs. GC	PFS	Advanced CCA with *FGFR2* rearrangements	PFS = 7.0 for patients with *FGFR2* fusions/rearrangements.	[43]
NCT01752920 (ARQ 087)	I/II (1st, 2nd, or later)	Derazantinib	Patients with AEs (2nd, ORR)	29 unresectable iCCA patients with *FGFR2* fusion	ORR = 20.7%	[44]
NCT02989857 (ClarIDHy)	III (2nd or 3rd)	Ivosidenib vs. placebo	PFS	185 advanced CCA patients with mutant *IDH1*	PFS was significantly improved with ivosidenib.	[45]
CTRI/2019/05/019323I ^4^ (BEER BTC)	II/III (2nd)	Bevacizumab + erlotinib vs. active surveillance	PFS	98 BTC patients with disease stabilization after 6 months of gemcitabine-based therapy	Bevacizumab + erlotinib improved PFS.	[46]
NCT01206049 (GOC-B-P)	II (1st)	Chemotherapy ^6^ + panitumumab (Arm A) vs. chemotherapy + bevacizumab (Arm B)	PFS	88 BTC patients without *KRAS* exon 2 mutation	42% (arm A) and 53% (arm B) of the patients showed PFS at 6 months; the primary endpoint was not met.	[47]
NCT01389414 (Vecti-BIL study)	II (1st)	Panitumumab + GEMOX vs. GEMOX	PFS	89 advanced BCT patients with wild-type *KRAS*	No survival differences were observed between the two groups.	[48]
NCT00552149 (BINGO)	II (1st)	GEMOX with or without cetuximab	PFS	150 advanced BTCs patients	63% and 54% of the patients showed PFS > 4 months in GEMOX + cetuximab and GEMOX alone groups, respectively.	[49]
NCT01149122	III (1st)	GEMOX + erlotinib vs. GEMOX	PFS	268 metastatic BCT patients	No significant difference in PFS was noted between the two groups. Significantly more patients had an OR in the GEMOX + erlotinib group (*p* = 0.005).	[50]
NCT03212274 (NCI 10129 trial)	II (2nd or later)	Olaparib	ORR	30 CCA patients with mutant *IDH*	No objective responses were seen; 27% of the patients showed a PFS of ≥6 months.	[51]
NCT05506943 (COMPANION-002)	II (2nd)	CTX-009 + paclitaxel vs. paclitaxel	ORR	150 advanced BTC patients	CTX-009 with paclitaxel showed an ORR = 37.5%.	[52]
NCT02034110 (ROAR trial)	II (Patients with no standard treatment options)	Dabrafenib + trametinib	ORR	43 advanced BTC patients with *BRAFV600E*-mutation	ORR = 53%.	[53]
NCT00939848 (ABC-03)	II (1st)	GC vs. GC + cediranib	PFS	124 advanced BTCs patients	PFS = 8.0 months in the cediranib group, 7.4 months in the GC-alone group.	[54]
jRCT2091220423 ^5^ (HERB; NCCH1805)	II (2nd or later)	Trastuzumab–Deruxtecan	ORR	32 unresectable or recurrent BTC with *HER2-*positive	ORR = 36.4% for *HER2*-positive disease. ORR = 12.5% for *HER2*-low disease.	[55]
NCT01953926 (SUMMIT)	II (No treatment with any HER2-directed tyrosine kinase inhibitor)	Neratinib	ORR	25 treatment-refractory BTC patients with *HER2* mutations	ORR = 16%.	[56]
NCT04466891 (HERIZON-BTC-01)	II (2nd or later)	Zanidatamab	ORR by IHC	87 BTC patients with *HER2*-amplified, advanced disease.	ORR= 41 · 3%.	[57]
NCT02091141 (MyPathway)	II (2nd)	Pertuzumab + trastuzumab	ORR	39 treated BTC patients with *HER2* amplification/overexpression	ORR = 23%.	[58]

^1^ Trial IC registered in ClinicalTrials.gov. ^2^ Targets of these agents are as follows: erdafitinib; pan-FGFR inhibitor, futibatinib; a covalent FGFR inhibitor, pemigatinib: FGFR1-3 inhibitor, infigratinib; FGFR 1-3 inhibitor, derazantinib; multi-kinase inhibitor with potent pan-FGFR activity, ivosidenib; IDH1 inhibitor, bevacizumab; VEGF-A antibody, erlotinib; EGFR inhibitor, panitumumab; EGFR antibody, cetuximab; EGFR antibody, olaparib; poly ADP ribose polymerase inhibitor, CTX-009; a bispecific antibody targeting both DLL4 and VEGF-A, dabrafenib; BRAF kinase inhibitor, trametinib; MEK inhibitor, cediranib; pan-VEGFR inhibitor, trastuzumab–deruxtecan; an antibody conjugate composed of anti-HER2 antibody, a cleavable tetrapeptide-based linker, and topoisomerase I inhibitor, neratinib; pan-HER inhibitor, zanidatamab; a bispecific antibody targeting two distinct HER2 epitopes, pertuzumab; anti-HER2 antibody, trastuzumab; anti-HER2 antibody. ^3^ PEP; primary endpoint. ^4^ CTRI; Clinical Trials Registry of India. ^5^ jRCTs; Japan Registry of Clinical Trials. ^6^ Chemotherapy; Patients received gemcitabine, oxaliplatin, and capecitabine as a chemotherapy. Abbreviations: ORR; objective response rate, CCA; cholangiocarcinoma, iCCA; intra hepatic cholangiocarcinoma, CR; complete response, PR; partial response, PFS; progression-free survival, AE; adverse events, BTC; biliary tract cancers, GEMOX; gemcitabine and oxaliplatin, GC; gemcitabine + cisplatin, IHC; immunohistochemistry.

**Table 3 cancers-17-01243-t003:** Phase II/III clinical trials using immune checkpoint inhibitors for cholangiocarcinoma.

Trial ID ^1^ (Study Name)	Phase (Line)	Agents	PEP ^2^	Patients	Results	Ref.
NCT03875235 (TOPAZ-1)	III (1st)	Durvalumab + GC vs. GC	OS	685 unresectable BTC patients	Durvalumab + GC showed robust and sustained OS benefits.	[134]
NCT02628067 (KEYNOTE-158)	II (2nd or later)	Pembrolizumab	ORR	63 advanced BTC patients	ORR = 0% in TMB-high group, 9% in non-TMB-high group.	[65]
NCT04003636 (KEYNOTE-966)	III (1st)	Pembrolizumab + GC vs. GC	OS	1069 unresectable BTC patients	Significant improvement in OS compared with GC.	[135]
NCT03951597 (JS001)	II ^3^ (1st)	Toripalimab, lenvatinib, and GEMOX	ORR	30 unresectable iCCA patients	ORR = 80%. Twenty-three achieved PR, and one achieved CR.	[136]

^1^ Trial IC registered in ClinicalTrials.gov. ^2^ PEP; primary endpoint. ^3^ This trial is conducted on patients with no previous ICI therapy. Abbreviations: GC: gemcitabine and cisplatin, OS: overall survival, BTC: biliary tract cancer, ORR: objective response rate, TMB: tumor mutation burden, GEMOX: gemcitabine and oxaliplatin, iCCA: intrahepatic cholangiocarcinoma, PR: partial response, CR: complete response.

## Data Availability

Data in this review paper are openly available from references.

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
