# Peer review of "Biomarkers and Management of Cholangiocarcinoma: Unveiling New Horizons for Precision Therapy"

_cancers, 2025, doi:10.3390/cancers17071243_

Round 1

Reviewer 1 Report

Comments and Suggestions for Authors

Manuscript entitled "Biomarkers for the Management of Cholangiocarcinoma: Unveiling New Horizons for Precision Therapy"

Major issues:

1. This work is not clinically relevant. The authors should summarize the most updated therapeutic strategies (agents) associated with the biology of cholangiocarcinoma. The most updated clinical trials should be summarized.

2. The authors should introduce the molecular aberrations and (molecular/pathologic) classifications and etiologies of cholangiocarcinoma.

3. More immunotherapy-related cancer biology and treatments should be mentioned.

4. Overall, this work is not solid enough to be published here.

Comments on the Quality of English Language

Acceptable.

Author Response

We sincerely appreciate the reviewers’ valuable comments and suggestions, which have greatly contributed to improving the quality of our manuscript. We fully agree with the reviewers’ points and have carefully revised the manuscript accordingly. Below, we outline the specific changes we have made in response to their feedback.

Reviewer 1

Inquiry 1.

This work is not clinically relevant. The authors should summarize the most updated therapeutic strategies (agents) associated with the biology of cholangiocarcinoma. The most updated clinical trials should be summarized.

Reply 1.

I greatly appreciate the careful review and constructive feedback. I concur with their assessments and have made the necessary revisions to address their concerns.

Based on the comment, I have summarized the most updated therapeutic strategies (agents) associated with the biology of cholangiocarcinoma in Table 1 (Phase II/III clinical trials of chemotherapy for cholangiocarcinoma), Table 2 (Phase II/III clinical trials using molecular targeted agents for cholangiocarcinoma), and Table 3 (Table 3. Phase II/III clinical trials using immune checkpoint inhibitors for cholangiocarcinoma). I also added several references for new tables.

Inquiry 2.

The authors should introduce the molecular aberrations and (molecular/pathologic) classifications and etiologies of cholangiocarcinoma.

Reply 2.

We would like to express our gratitude to the reviewers for their thoughtful suggestions. We agreed with their observations and made the revise as follows.

The new statements regarding the molecular aberrations and (molecular/pathologic) classifications and etiologies in page 6, line 164-166, page 6 line 172-page 7 line 210, including paragraphs for "3.1 Etiologies, pathologies and genetic alterations in the small-duct type of intrahepatic cholangiocarcinoma" and "3.2 Etiologies, pathologies and genetic alterations in the large-duct type of intrahepatic, perihilar, and distal cholangiocarcinoma".

Inquiry 3.

More immunotherapy-related cancer biology and treatments should be mentioned.

Replay 3.

Thank you for the valuable suggestion. I have presented the new table as "Table 3" for the summary of immunotherapy-related cancer biology and treatments. I also added the schematic representation of the tumor immune microenvironment in CCA as Figure 3. The new statement regarding this revise is shown in page 18, line 594 - 618, and Page 19, line 645-647.

Inquiry 4.

Overall, this work is not solid enough to be published here.

Reply 4.

I sincerely appreciate the reviewer’s time and effort in evaluating our manuscript. I would like to clarify that this is a review article, and as such, it is based on previously published studies rather than new hypotheses. We have made every effort to ensure that our discussion remains grounded in established research and have minimized speculative statements. Therefore, we believe that the content is sufficiently solid.

However, if certain aspects of our manuscript appear insufficiently substantiated, we would be grateful if the reviewer could specify which sections are considered “not solid enough to be published here.” We are fully prepared to further revise the manuscript to address any specific concerns.

Reviewer 2 Report

Comments and Suggestions for Authors

The review of Nishida N. is clear and well written and the topic is of great interest, since cholangiocarcinoma is a tumor orphan of good specific biomarkers.

The various categories of biomarkers are treated quite thoroughly, so the review is quite complete from this point of view.

However, I must point out that paragraph 6 seems a bit off-topic compared to everything else, as it is not possible to clearly understand the connection between microenvironment and biomarkers, especially considering the fact that biomarkers should then be easily usable in clinical practice. I suggest to extensively modifying the paragraph or removing it altogether.

Finally, the resolution of the figures should be increased/improved.

Author Response

We sincerely appreciate the reviewers’ valuable comments and suggestions, which have greatly contributed to improving the quality of our manuscript. We fully agree with the reviewers’ points and have carefully revised the manuscript accordingly. Below, we outline the specific changes we have made in response to their feedback.

Reviewer 2

Inquiry

The review of Nishida N. is clear and well written and the topic is of great interest, since cholangiocarcinoma is a tumor orphan of good specific biomarkers. The various categories of biomarkers are treated quite thoroughly, so the review is quite complete from this point of view.

However, I must point out that paragraph 6 seems a bit off-topic compared to everything else, as it is not possible to clearly understand the connection between microenvironment and biomarkers, especially considering the fact that biomarkers should then be easily usable in clinical practice. I suggest to extensively modifying the paragraph or removing it altogether. Finally, the resolution of the figures should be increased/improved.

Reply

We sincerely appreciate the reviewer’s insightful and positive comments. We acknowledge that, in the context of biomarkers for cholangiocarcinoma (CCA), the original Paragraph 6 may have been somewhat off-topic compared to other sections, and we recognize that the connection between the tumor microenvironment and biomarkers may not have been clearly articulated.

However, given that immune checkpoint inhibitors (ICIs) are increasingly being used in clinical practice for CCA, we believe that understanding the tumor immune microenvironment specific to CCA will play a crucial role in future biomarker development. Additionally, in response to Reviewer 1’s comment that “More immunotherapy-related cancer biology and treatments should be mentioned,” we have incorporated a summary of immunotherapy-related cancer biology and treatments in Table 3 and included Figure 3, titled “Schematic representation of the tumor immune microenvironment in cholangiocarcinoma”, to enhance the understanding of biomarker exploration from the perspective of the tumor microenvironment.

At the same time, we acknowledge that further accumulation of clinical data on ICI-treated cases will be necessary for the development of reliable biomarkers in this context. Therefore, we have revised the title to "Biomarkers and Management of Cholangiocarcinoma" to reflect not only the discussion on biomarkers but also broader aspects of CCA management.

Additionally, we have improved the resolution of the figures to enhance clarity.

Reviewer 3 Report

Comments and Suggestions for Authors

The present review summarized the development of biomarker in cholangiocarcinoma (CCA), Conventional tumor markers lack diagnostic accuracy; recent advancements in next generation sequencing have identified actionable mutations, such as FGFR2 fusions and IDH1/2 mutations, enabling targeted therapies that improve survival. Cancer panels facilitating multiplex profiling reveal actionable mutations, with mutation-guided therapies markedly enhancing outcomes. Liquid biopsies, including cfDNA and ctDNA, offer non-invasive, real-time tumor monitoring. Non-coding RNAs in serum and bile also show promise as diagnostic and prognostic biomarkers. The review discussed the current state of biomarker development for CCA, encompassing their utility in diagnosis, prognosis, and therapeutic stratification. The review also explored emerging trends and challenges in integrating biomarkers into clinical practice, highlighting their potential to shape the future of precision medicine for CCA treatmentThe review is overall well organized and written. Some minor points are listed as below.

1. Subtitles can be added for the serum and plasma biomarkers part and genetic aberrations just as the following parts.

2. Figures showing the biomarkers for tumor immune microenvironment can be supplemented.

Comments on the Quality of English Language

The English could be improved to more clearly express the research.

Author Response

We sincerely appreciate the reviewers’ valuable comments and suggestions, which have greatly contributed to improving the quality of our manuscript. We fully agree with the reviewers’ points and have carefully revised the manuscript accordingly. Below, we outline the specific changes we have made in response to their feedback.

Reviewer 3

The present review summarized the development of biomarker in cholangiocarcinoma (CCA), Conventional tumor markers lack diagnostic accuracy; recent advancements in next generation sequencing have identified actionable mutations, such as FGFR2 fusions and IDH1/2 mutations, enabling targeted therapies that improve survival. Cancer panels facilitating multiplex profiling reveal actionable mutations, with mutation-guided therapies markedly enhancing outcomes. Liquid biopsies, including cfDNA and ctDNA, offer non-invasive, real-time tumor monitoring. Non-coding RNAs in serum and bile also show promise as diagnostic and prognostic biomarkers. The review discussed the current state of biomarker development for CCA, encompassing their utility in diagnosis, prognosis, and therapeutic stratification. The review also explored emerging trends and challenges in integrating biomarkers into clinical practice, highlighting their potential to shape the future of precision medicine for CCA treatment The review is overall well organized and written. Some minor points are listed as below.

Reply

We would like to express our sincere gratitude to the reviewers for their thorough evaluation, thoughtful suggestions, and positive comments. I believe that the comments have significantly contributed to strengthening our manuscript.

Inquiry 1.

Subtitles can be added for the serum and plasma biomarkers part and genetic aberrations just as the following parts.

Reply 1.

I have added the subtitles in the serum and plasma biomarkers part and genetic aberrations, as follows.

2. Serum and plasma biomarkers for cholangiocarcinoma

2.1 Conventional serum and plasma biomarkers

2.2 Serum and plasma biomarkers under development

2.3 Cancer-associated metabolite

3. Genetic aberrations in cholangiocarcinoma

3.1 Etiologies, pathologies and genetic alterations in the small-duct type of intrahepatic cholangiocarcinoma

3.2 Etiologies, pathologies and genetic alterations in the large-duct type of intrahepatic, perihilar, and distal cholangiocarcinoma

3.3 Cancer panels and companion diagnostics for cholangiocarcinoma

Inquiry 2.

Figures showing the biomarkers for tumor immune microenvironment can be supplemented.

The English could be improved to more clearly express the research.

Reply 2.

I have added Figure 3, titled “Schematic representation of the tumor immune microenvironment in cholangiocarcinoma”, to enhance the understanding of biomarker exploration from the perspective of the tumor microenvironment.

I submitted the manuscript for professional English editing and have reviewed the text again based on the provided corrections.

Round 2

Reviewer 1 Report

Comments and Suggestions for Authors

The revision is acceptable in the present form.

Comments on the Quality of English Language

acceptable.

Reviewer 2 Report

Comments and Suggestions for Authors

Thanks to the authors to the revised version of the manuscript that includes all the modifications suggested. For me is now acceptable for publication.